# Anticancer and Structure Activity Relationship of Non-Symmetrical Choline Kinase Inhibitors

**DOI:** 10.3390/pharmaceutics13091360

**Published:** 2021-08-29

**Authors:** Santiago Schiaffino-Ortega, Elena Mariotto, Pilar María Luque-Navarro, María Kimatrai-Salvador, Pablo Rios-Marco, Ramon Hurtado-Guerrero, Carmen Marco, María Paz Carrasco-Jimenez, Giampietro Viola, Luisa Carlota López-Cara

**Affiliations:** 1Department of Pharmaceutical and Organic Chemistry, Faculty of Pharmacy, Campus of Cartuja, University of Granada, 18071 Granada, Spain; schaiffino@hotmail.com (S.S.-O.); pilarluque@ugr.es (P.M.L.-N.); mkimatrai@gmail.com (M.K.-S.); 2Laboratory of Oncohematology, Department of Woman’s and Child’s Health, University of Padova, 35128 Padova, Italy; elena.mariotto@unipd.it; 3Department of Biochemistry and Molecular Biology I, Faculty of Sciences, University of Granada, 18071 Granada, Spain; priosm@ugr.es (P.R.-M.); cmarco@ugr.es (C.M.); 4Institute of Biocomputation and Physics of Complex Systems (BIFI), Mariano Esquillor s/n, Campus Rio Ebro, Edificio I+D, University of Zaragoza, 50018 Zaragoza, Spain; rhurtado@bifi.es

**Keywords:** antitumoral drug, choline kinase inhibition, choline uptake

## Abstract

Choline kinase inhibitors are an outstanding class of cytotoxic compounds useful for the treatment of different forms of cancer since aberrant choline metabolism is a feature of neoplastic cells. Here, we present the most in-depth structure-activity relationship studies of an interesting series of non-symmetric choline kinase inhibitors previously reported by our group: **3a**–**h** and **4a**–**h**. They are characterized by cationic heads of 3-aminophenol bound to 4-(dimethylamino)- or 4-(pyrrolidin-1-yl)pyridinium through several linkers. These derivatives were evaluated both for their inhibitory activity on the enzyme and their antiproliferative activity in a panel of six human tumor cell lines. The compounds with the N-atom connected to the linker (**4a**–**h**) show the best inhibitory results, in the manner of results supported by docking studies. On the contrary, the best antiproliferative compounds were those with the O-atom bounded to the linker (**3a**–**h**). On the other hand, as was predictable in both families, the inhibitory effect on the enzyme is better the shorter the length of the linker. However, in tumor cells, lipophilicity and choline uptake inhibition could play a decisive role. Interestingly, compounds **3c** and **4f**, selected for both their ability to inhibit the enzyme and good antiproliferative activity, are endowed with low toxicity in non-tumoral cells (e.g., human peripheral lymphocytes) concerning cancer cells. These compounds were also able to induce apoptosis in Jurkat leukemic cells without causing significant variations of the cell cycle. It is worth mentioning that these derivatives, besides their inhibitory effect on choline kinase, displayed a modest ability to inhibit choline uptake thus suggesting that this mechanism may also contribute to the observed cytotoxicity.

## 1. Introduction

Cancer is among the main world causes of mortality and morbidity. That is in part due to the resistance to drugs given by the tumor microenvironment [1]. For this reason, despite being an intensively investigated disease, new drugs and therapeutic strategies are still urgently needed to fight it more efficiently. Currently, many of these new therapeutic strategies are focused on nanotechnology to improve targeting [2,3,4].

Membranes in eukaryotic cells are composed principally of phospholipids and the zwitterionic glycerophospholipids. Phosphatidylcholine (PC) and phosphatidylethanolamine (PE) are the most common phospholipids species in this type of cell. As well as in cell membranes, PC constitutes the main phospholipid class contained in lipoproteins, biliary lipids aggregates, and lung surfactants. [5,6] It serves as a direct substrate for sphingomyelin (SM) synthesis and it is also a fundamental source of the second messenger diacylglycerol (DAG), phosphatidic acid, lysophosphatidic acid, and arachidonic acid that can be further metabolized to other signalling molecules. PC was also reported as a mitogen required for DNA synthesis, induced by growth factors [5]. PC can be synthesized from the quaternary amine choline via the de-novo pathway, also known as the Kennedy pathway and is likewise responsible for the synthesis of PE from ethanolamine. The 95% of the total pool of choline located in most tissues will enter this metabolic route [5] whose first step is catalyzed by the choline kinase (ChoK) enzyme. ChoK is a cytosolic enzyme that catalyzes the ATP-dependent phosphorylation of choline in the presence of Mg++ [5,7].

The choline kinase family in humans comprises two chok genes: Chok-α and chok-β genes, that encode three isoforms; ChoKα1 (457aa, 52 kDa), Chokα2 (439aa, 50 kDa), and ChoKβ (395aa, 45 kDa). While ChoKα1 and ChoKα2 derived from the same gene by differential splicing and are almost identical (except that ChoKα1 has 18 amino acids extra), the homology between ChoKα and ChoKβ is 60%, being higher in the choline domain [7,8]. 

Normal cell proliferation is characterized by the strict duplication of all cellular components in a stringent temporal order. In the particular case of lipids, the maintenance of the status of PC as one of the structural elements of cell membranes and source of important mitotic second messengers would have important implications in cell proliferation and apoptosis [4,9]. Actually, among other changes, tumors display elevated levels of phospholipids characterized by the increase of phosphocholine (PCho) and total choline-containing metabolites. [10,11,12] ChoKα appears overexpressed in 40–60% of breast, prostate, lung, colon, ovarian, and bladder cancers, which represent 70% of all cancers in developed countries. For this reason, ChoK has been proposed as a prognostic marker of progression [7] as well as a molecular target in oncology [13,14,15,16,17]. 

Due to its vital and widely studied role in cell division and tumor formation, ChoK emerged as a potential target for various cancers [11,12,13], particularly, Ras-induced carcinogenesis. The Ras effectors serine/threonine kinase (Raf-1), the Ral-GDP dissociation stimulator (Ral-GDS) and the phosphatidylinositol 3-kinase (PI3K) are all involved in ChoK activation during tumorigenesis [7,9,18].

We have previously reported the synthesis and the biological evaluation of a new family of non-symmetrical monocationic compounds (Figure 1) structurally related to hemicolinium-3 (**HC-3**), endowed with antitumor activity including a 3-amino-phenol moiety bound to 4-pyrrolidinopyridinium or 4-dimethylaminopyridinium groups through several linkers [19]. In the last decade, numerous studies on ChoKα1 inhibitors were published, including their great applicability in different diseases. For this reason, we have been encouraged to carry out further studies on our library of compounds **3a**–**h** and **4a**–**h** [19].

The residues that constitute the ATP binding site are quite different to those that form the Cho binding site in the enzyme. This suggested the idea of synthesizing non-symmetric monocationic inhibitors with one cationic head that could be inserted into the Cho binding site and another fragment that could mimic the ATP adenine moiety. The size of symmetrical biscationic inhibitors was appropriate to bind simultaneously in both the ATP and Cho putative binding sites of the protein model [19,20,21,22,23,24]. On the other hand, we have recently reported a new series of small monocationic molecules where the inhibition of choline uptake has emerged as a major contributor to the antiproliferative outcome of this class of compounds [13]. The results provided in the present study can complement the earlier outcomes reported since docking studies have been done in more appropriate crystal structures. The inhibitory activity has been described over the isolate ChoKα1 enzyme and antiproliferative effects against a panel of six human cancer cell lines have been tested to complete and validate the earlier experiments reported. The results of this study would provide additional information about the molecule’s mechanism of action and the design requirements of new structures.

## 2. Materials and Methods

### 2.1. Chemistry

The compounds **3a**–**h** and **4a**–**h** were synthesized following the protocols described previously [19].

### 2.2. Cloning, Purification of CK, and Inhibition of Choline Kinase α1 by Compounds ***3a**–**h*** and ***4a**–**h***

Details about cloning and purification of human ChoKα1 and ChoKβ have been previously reported [17]. 

The effect of compounds **3a**–**h** and **4a**–**h** on ChoKα1 was assayed in purified ChoKα1 as previously described [21,23,24], by determining the rate of incorporation of ^14^C from [methyl-^14^C]choline into phosphocholine, both in the absence (control) or presence of different inhibitor concentrations. Briefly, 20 ng of purified ChoKα1 were incubated with 1 mM [methyl-^14^C]choline (4500 dpm/nmol) in 100 mM Tris-HCl (pH 8.5), 10 mM MgCl_2_, 10 mM ATP, and incubated at 37 °C for 10 min. The reaction was stopped by immersing the reaction tubes in boiling water for 3 min. Aliquots of the reaction were applied to the origin of silica gel plates in the presence of phosphocholine (0.1 mg) and choline (0.1 mg) as carriers. The chromatography was developed in methanol/0.6% NaCl/28% NH_4_OH in water (50:50:5, *v*/*v*/*v*) as a solvent, and phosphocholine was visualized under exposure to iodine vapor. The corresponding spot was scraped and transferred to scintillation vials for measurement of radioactivity by a Beckman 6000-TA (Madrid, Spain) liquid-scintillation counter. The 50% inhibitory concentrations (IC_50_ value) were determined from the % activity of the enzyme at different concentrations of synthetic inhibitor using a sigmoidal dose-response curve (the ED_50_plus v1.0 software).

### 2.3. Molecular-Modeling Studies

Molecular-modeling studies were performed using the Sybyl program [25]. Crystal structures of human ChoKα1 in complex with compounds **5** (PDB entry 3ZM9) and **6** (PDB entry 4BR3) were used for docking studies. In both cases, using the Structure Preparation Tool module of Sybyl refined the protein structure. Missing side chains of those residues situated far away from the binding sites were added and protein N-terminal and C-terminal were fixed with ACE and NME, respectively. Hydrogens and charges were also added and the protonation type of Glu, Asp, Gln, and Asp was analyzed and fixed. Hydrogen orientations were also checked in order to maintain intramolecular hydrogen bonds within the protein. Structures of compounds **3a**–**h** and **4a**–**h** were constructed from standard fragments of the Libraries of the Sybyl program, and used as ligands for docking studies. As previously described [26], a new type of atom was necessary to define in order to build the molecules: N.ar4, the quaternary nitrogen of the pyridinium fragments. Additional parameters were also developed from ab initio calculations to optimize the geometry of these molecules. Atomic charges were calculated by means of Gaussian Program [27] and optimizations were undertaken using the BFGS method. The Surflex-Dock [28] module implemented in the Sybyl program was used for docking studies. Surflex Dock Protomol was prepared using compound **5** or **6** inserted into the ChoK binding site, with a threshold value of 0.5 and a Bloat of 0 A. Surflex-Dock GeomX (SFXC) protocol was used, the search grid was expanded in 5 Å, 50 additional starting conformations were used for each molecule and 30 conformations per fragment. The results were analyzed using the Sybyl program and the most stable pose for each molecule was chosen as the preferred one inside the ChoK enzyme. Figures were built using the PyMOL program [29].

### 2.4. Antiproliferative Assays in Cancer Cells

Human cervix carcinoma (HeLa) and human breast cancer (MCF-7) were grown in the DMEM medium (Gibco, Milan, Italy). B-acute lymphoblastic leukemia (RS4;11), T-acute lymphoblastic leukemia (CCRF-CEM and Jurkat), human promyelocytic cells (HL-60), and human colon adenocarcinoma (HT-29) cells were grown in RPMI medium (Gibco, Milan, Italy). Both media were supplemented with 115 units/mL of penicillin G (Gibco, Milan, Italy), 115 μg/mL of streptomycin (Invitrogen, Milan, Italy), and 10% FBS (Invitrogen, Milan, Italy). Cell lines were tested for mycoplasma contamination every 6 months by the RT-PCR analysis. Stock solutions (10 mM) of the different compounds were obtained by dissolving them in DMSO. Individual wells of 96-well tissue-culture microtiter plates were inoculated with 100 μL of complete medium containing 5 × 10^3^ adherent cells or 25 × 10^3^ leukemia cells. The plates were incubated at 37 °C in a humidified 5% CO_2_ incubator for 18 h prior to the experiments. After medium removal, 100 μL of fresh medium containing the test compound at different concentrations was added to each well and incubated at 37 °C for 72 h. Cell viability was assayed by the (3-(4,5-dimethylthiazol-2-yl)-2,5-diphenyl tetrazolium bromide (MTT) test as previously described [3]. The GI_50_ was defined as the compound concentration required to inhibit cell proliferation by 50%, in comparison with cells treated with the maximum amount of DMSO (0.25%) and considered as 100% viability.

### 2.5. Antiproliferative Activity in Peripheral Blood Lymphocytes (PBL)

Additional experiments were conducted with peripheral blood lymphocytes (PBL) from healthy donors obtained as described previously [30]. For cytotoxicity evaluations in proliferating PBL cultures, non-adherent cells were resuspended at 5 × 10^5^ cells/mL in a growth medium containing 2.5 µg/mL PHA (Irvine Scientific). Different concentrations of the test compounds were added, and viability was determined 72 h later by the MTT test. For cytotoxicity evaluations in resting PBL cultures, non-adherent cells were resuspended (5 × 10^5^ cells/mL) and treated for 72 h with the test compounds, as described above.

### 2.6. Cell Cycle Analysis

Jurkat cells were treated with the test compounds for 24 h. Cells were harvested by centrifugation and fixed with 70% (*v*/*v*) cold ethanol. Cells were lysed with 0.1% (*v*/*v*) Triton X-100 containing RNase A and stained with PI. A Beckman Coulter Cytomics FC500 instrument and MultiCycle for Windows software from Phoenix Flow Systems were used to analyze the cells.

### 2.7. Measurement of Apoptosis by Flow Cytometry

Jurkat cells were treated with the test compounds and after different times stained with both PI, to stain DNA, and annexin V-fluorescein isothiocyanate, to stain membrane PS exposed on the cell surface, following the instructions of the manufacturer (Roche Diagnostics) of the Annexin-V-Fluos reagent.

### 2.8. Choline Uptake Assay

Choline uptake was determined as previously reported [13]. Briefly, HepG2 cells (200,000 cells/well) were incubated for 10 min at 37 °C either in a medium containing different concentrations of ChoKα1 inhibitors or only with the medium as controls. The medium was then removed and the cells were immediately exposed to a medium containing [methyl-^14^C]choline (16 μM, 31 Ci/mol) for 10 min at 37 °C. The incorporation of choline was stopped by medium aspiration followed by two washes with ice-cold PBS containing 580 μM choline. The cells were solubilized in NaOH 0.1 N and an aliquot was used to determine the total amount of radiolabel taken up by the cells.

## 3. Results and Discussion

### 3.1. Preliminary Docking Studies

Human choline kinase α1 (hChoKα1; PDB id: 3G15) was chosen for the docking studies due to several reasons: (i) When 3D structures of both ChoKα1 (PDB ID: 3G15) and ChoKβ (PDB ID: 3FEG) isoforms were crystalized in complex with the first known inhibitor of ChoK, HC-3 suffered a phosphorylation on the morpholinium moiety inserted into the choline binding site of the ChoKβ isoform, thus the obtained crystal structure of the complex was constituted by ChoKβ, ADP, and phosphohemicolinium-3 (PHC-3). This situation did not occur with ChoKα, where it was possible to study the enzyme co-crystallized in complex with both, not phosphorylated HC-3 and ADP. Therefore, (ii) ChoKα1 co-crystallized with both ADP and HC-3 is more appropriated for docking studies in both binding sites than ChoKβ; and (iii) the described biological results indicate that ChoKα1 and not ChoKβ is the most suitable target to study and design new anticancer drugs [12]. We published the crystal structure of the ChoKα1 isoenzyme with two inhibitors characterized by a cationic head; the first one named Compound **5**: 1-[4-(4-{4-[(6-amino-9H-purin-9-yl) methyl] phenyl} butyl)benzyl]-4-(dimethylamino)-pyridinium (PDB ID: 3ZM9; see Figure 2A) [20,21] and a second one called Compound **6**: 1-(4-{4-[(6-amino-3H-purin-3-yl)methyl]phenyl}benzyl)-4-(dimethylamino)-pyridinium (PDB ID: 4BR3; see Figure 2B) [21]. The resemblances between compounds **5** and **6** with the compounds **3a**–**h** and **4a**–**h** indicate the use of these novel crystal structures obtained rather than the one provided with HC-3 to perform docking studies, given that these **3a**–**h** and **4a**–**h** structures are characterized in a similar manner by one cationic head (Figure 3, Appendix A).

### 3.2. Inhibition of ChoKα1 by Compounds ***3a**–**h*** and ***4a**–**h*** as well as Docking Studies

Table 1 summarizes the clogP, the inhibitory effect on purified human ChoKα 1 activity and the growth inhibitory effects against a panel of six different human tumor-cell lines. Of all the compounds tested, in general terms, those where the aminophenol system is connected by the N-atom (**4a**–**h**) are the ones that offer the best results in terms of enzyme inhibition. Among them, compound **4f** stands out with an IC_50_: 0.99 μM.

As for the length of the spacer, it does not seem to affect the enzyme inhibition. However, a significant increase of the inhibitory activity is observed in compounds where the spacer is longer and is connected by the O-atom to the aminophenol group, see compounds (**3a**–**b** vs. **3c**–**d** and **3e**–**f** vs. **3g**–**h**). This difference is not apparent when the compounds are connected to the aminophenol group by N-atom (**4a**–**h**), where the spacer does not seem to exert much influence. On the other hand, the activity is favored by the influence of the substituent in position 4 of the pyridinium ring. Thus, compounds substituted with dimethylamino as (**3e**–**h**) and (**4e**–**h**) show an appreciable improvement of the inhibition values. As explained in Section 3.1, the obtained crystal structures of the ChoKα1 isoenzyme with monocationic compounds **5** and **6** were considered more appropriate for docking studies of compounds **3a**–**h** and **4a**–**h**, which only have a cationic head than the crystal structure of the enzyme with **HC-3**.

Figure 2 shows a representation of the interactions of molecules **5** and **6** individually with the enzyme. The higher compound (PDB ID: 3ZM9, previously called number **5**, see Figure 2A) [20] has an adenine moiety and a 4-(dimethylamino)pyridinium cationic head, connected by a long linker [1,4-diphenylbutane]. This compound occupies a long active site from the ATP to the choline (Cho) binding sites (Figure 2A). The linker is connected to the N-9 adenine atom similarly to the ribose-adenine connection in ATP, and for this reason the adenine moiety can mimic the connection of the ATP cofactor to ChoK, being inserted into the ATP binding site [20]. The adenine moiety is stabilized by means of hydrophobic interactions with Leu144, Phe208, Ile209, and Leu313; and by two H-bonds with Glu207 and Ile209. The benzyl fragment connected to the adenine moiety is also stabilized by hydrophobic interactions with Arg117, Arg213, and Leu124 side chains. Finally, the 4-(dimethylamino)pyridinium fragment of this compound **5** is inserted into the ATP binding site and stabilized by π-cations interactions with Tyr333, Tyr354, Trp420, Tyr423, and Trp440.

Compound number **6** (PDB ID: 4BR3, see Figure 2B) has also an adenine moiety and a 4-(dimethylamino)pyridinium fragment. The linker is smaller being connected to the N-3 adenine atom, and for these reasons, one molecule of this compound cannot occupy simultaneously both ATP and Cho binding sites similarly to compound **5** [21]. In the crystal structure (Figure 2B) a partial density corresponding to the 3-benzyl adenine fragment of this compound (carbon atoms in green colors) was detected into the ATP binding site, and another whole molecule (carbon atoms in yellow colors) was observed into the Cho binding site. The adenine moiety inserted into the ATP binding site is also stabilized by hydrophobic interactions with Leu144, Phe208, Ile209, and Leu313; and by H-bonds with Glu207, Ile209, similarly to compound **5**. An additional H-bond between the adenine N-9 atom and the Arg213 side chain is also observed due to the high flexibility of this amino acid. The rest of this compound **6** which were inserted in the ATP binding site were not detected probably due to two reasons: i) The 1-benzyl-4-(dimethylamino)pyridinium fragment is situated outside of the protein, showing high flexibility and a really poor density and ii) the interaction of compound **6** with this region of the protein is not efficient, as was described recently [20]. Finally, molecule **6** (carbon atoms in yellow colors) inserted into the Cho binding site interacts with the protein very efficiently, being stabilized by π-cations interactions with Tyr333, Tyr354, Tyr440, Trp420, Trp423, Trp435, and Phe435. In particular, the biphenyl group shows optimal hydrophobic stacking interactions with Tyr354, and the 4-(dimethylamino)pyridinium moiety interacts through parallel π-cation interactions with Trp420. This new orientation of compound **6** inside the Cho binding site is accommodated by a conformational change of Tyr333 that have moved back and made an extra space in relation to compound **5**. The adenine fragment of this molecule inserted into the Cho binding site is outside of the enzyme and it does not show an interaction with the protein (see Appendix A), being the 1-(biphenyl-4-ylmethyl)-4-(dimethylamino)pyridinium the key fragment of this compound for the interaction into the ATP binding site [21].

Docking studies have been performed in both crystal structures and the analysis of the obtained poses indicates which compounds could be similar to compound **5** or compound **6**. In fact, compounds **3a**, **3b**, **3e**, **3f**, **4a**, **4b**, **4e**, and **4f** (Figure 3 and Appendix A of the Supporting Information) have shown good poses in the crystal structure of compound 6, and the correct poses of compounds **3c**, **3d**, **3g**, **3h**, **4c**, **4d**, **4g**, and **4h** (Figure 3 and Appendix A of the Supporting Information) were obtained in the crystal structure of compound **5**. The reason for these poses is the length of the linker. Furthermore, all compounds that have a benzene or biphenyl linker are inserted into the ChoK binding site similarly to compound **6**, and compounds that have a 1,2-diphenylethane or a 1,4-diphenylbutane linker are inserted into the ATP and ChoK binding sites, similarly to compound **5**.

As an example, Figure 3 (Panels A and B) shows the obtained poses of compounds **3f** (carbon atoms in white color) and **4f** (carbon atoms in orange color). Both compounds have a 4-(dimethylamino)pyridinium cationic head and a biphenyl linker. The cationic head in these compounds is also stabilized by π-cation interaction and the biphenyl group shows optimal hydrophobic stacking interactions, being very similar to compound **6**. Nevertheless, in compound **3f** the linker is connected to the 3-aminophenol O-atom, and in compound **4f** the linker is connected to the 3-aminophenol N-atom. The 3-aminophenol fragment of compound **4f** is stabilized by two H-bonds, with Tyr354 and Glu434, while in compound **3f** is stabilized by only one H-bond with Tyr354, and for this reason, compound **4f** has lower IC_50_ (0.99 ± 0.17 μM) compared to **3f** (6.39 ± 0.46 μM) regarding the inhibition of ChoKα1. Concerning compounds **3a**, **3b**, **3e**, **4a**, **4b**, and **4e**, the obtained poses highlight that the increased size of the 4-(pyrrolidin-1-yl)pyridininium cationic head (**3a** and **3b**) negatively affects the π-cations interactions (Appendix A), resulting in a larger IC_50_ compared to the equivalent compounds with a 4-(dimethylamino)pyridinium (**3e**). On the other hand, the biphenyl linker connected to the 3-aminophenol N-atom (compound **4b**) allows two H-bonds with Tyr354 and Glu434 and hence increases the inhibitory activity. Figure 3 also shows the obtained pose of compounds **3g** (carbon atoms in cyan colors) and **4g** (carbon atoms in magenta color), which also confirm the previous hypotheses. With respect to the **3h** compound, it does not seem to be affected by the volume of the substituent in position 4 of the cationic head regarding the family containing the -O-atom of the aminophenol connected to the rest of the molecule, while in their isomers (**4d** vs. **4h**) an unexplained decrease of the enzymatic activity of **4h** is observed (Appendix A).

### 3.3. In Vitro Antiproliferative Activities

The derivative **3c** was the most active compound identified in this study, inhibiting the growth of HT-29, HeLa, MCF-7, CCRF-CEM, HL-60, and RS4;11cells with GI_50_ values ranging from 630 to 11 nM, resulting in 30-fold more potent than MN-48b [23] against HL-60 leukemic cells.

Concerning the effect of the rest of the final compounds, it can be observed that: The compounds that offer inhibition values at the nanomolar range correspond to **3c**–**d** and **4c**–**d**, i.e., those with the longest spacer (2 or 4 carbons). In addition, these compounds are also those that possess a pyrrolidine as a substituent in position 4 of the pyridinium ring. There are no major differences in activity between the compounds linked by oxygen and those linked by nitrogen, although the most active compound corresponds to **3c**. The fact that the compounds with the best IC_50_ values of enzyme inhibition do not correspond to those with the best antiproliferative values may be due to differences in the clogP, which in turn lead to an increase in membrane uptake. Indeed, **3c**–**d** and **4c**–**d** have higher clogP values than their counterparts with shorter spacers (**3a**–**b** and **4a**–**b**). On the other hand, compounds with dimethylamino substituents on the 4-position of the pyridinium ring show a significant reduction in enzyme inhibition in comparison to the pyrrolidine substituted ones which could be due to the larger volume of the 4-(pyrrolidin-1-yl)pyridinium cationic head that produces a lower π-cation interaction.

### 3.4. Effects of Compound ***3c*** and ***4f*** in Non-Tumor Cells

To investigate the cytotoxic potential of these compounds in normal human cells, the two compounds **3c** and **4f** were evaluated in vitro against peripheral blood lymphocytes (PBL) collected from healthy donors. Compound **3c** showed an GI_50_ greater than 10 µM, both in quiescent lymphocytes and in proliferating lymphocytes stimulated with phytohemagglutinin (PHA) (Table 2), suggesting that this compound specifically targets tumoral cells. On the other hand, compound **4f** exhibits a minimal toxicity with GI_50_ of 7.6 and 3.6 µM in quiescent and PHA-stimulated lymphocytes, respectively. Nevertheless, these values were almost 120 times higher than that observed against the T-lymphoblastic cell line (CCRF-CEM).

### 3.5. Cell Cycle Analysis

To study in detail the mechanism of action of these compounds, we used a T-acute lymphoblastic leukemia cell line (Jurkat) against which we demonstrated a particular efficacy of the ChoKα1 inhibitors [16]. We first studied the effects on cell cycle following treatment with the two compounds and the results are shown in Figure 4. As can be seen, both compounds cause only modest and not significative changes in the cell cycle even at the highest concentration used (5 µM).

### 3.6. Measurement of Apoptosis by Flow Cytometry

To better depict the cytotoxic effect of compounds **3c** and **4f**, Jurkat cells were subjected to double labeling with annexin-V and PI and then analyzed by flow cytometry. This procedure allows the quantitation of four different cell populations: Living cells (annexin-V^−^/PI^−^), early apoptotic cells (annexin-V^+^/PI^−^), late apoptotic cells (annexin-V^+^/PI^+^), and necrotic cells (annexin-V^−^/PI^+^). Jurkat cells treated with **3c** and **4f** (Figure 5) displayed a significant increase of apoptotic cells after 24 h of treatment in a concentration dependent manner.

### 3.7. Choline Uptake Assay

Since recently, we have demonstrated that some ChoKα1 inhibitors, in addition to their effects on *h*ChoKα1 activity, are able to reduce choline uptake into the cell [13], we decided to investigate whether these compounds were also capable of inhibiting choline uptake in HepG2 cells. The two compounds chosen were **3d** (Table 1 and Table 3), which show moderate enzyme inhibition but very good antiproliferative values. On the other hand, the other compound chosen was **4f**, which has a good inhibitory IC_50_ but modest GI_50_ values on cell growth (Table 1 and Table 3). Compound **3d** inhibits choline uptake moderately, so its high antiproliferative activity cannot be directly attributed only to choline uptake but to a dual effect between choline uptake and enzyme inhibition. In contrast, the compound with poor effect on cell growth inhibition **4f**, has good enzyme and choline uptake inhibition values. In addition, **4f** shows low lipophilicity, which perhaps does not allow the compound to pass through the plasma membrane. Therefore, it follows that a good value for choline uptake and enzyme inhibition is not sufficient, and the need to control the lipophilicity of inhibitors to allow the molecules to pass through the plasma membrane comes into play in the equation.

## 4. Conclusions

The previously published compounds **3a**–**h** and **4a**–**h** have been exposed to more detailed studies which have revealed interesting results. The initial hypothesis suggested that the derivative compounds with a longer linker analyzed in this work, would potent hChoKα1 inhibitors since they could be inserted in both ATP and Cho binding sites of the hChoKα1 enzyme simultaneously. According to this premise, compounds **3c**, **3d**, **3g**, **3h**, **4c**, **4d**, **4g**, and **4h**, whose results were obtained after docking studies and the poses analyses indicated that they could be similar to the model compound so-called number **5** (PDB ID: 3ZM9), should be the most potent. However, compound **4f** (a compound that has shown good poses in the crystal structure of compound **6** (PDB ID: 4BR3), the one with a smaller linker) shows the best IC_50_ (0.99 ± 0.17 μM) for the inhibition of ChoKα1. This can be explained since not one but two molecules were found occupying the ATP and Cho binding sites all at once, one in the ATP binding site and another into the Cho binding site were detected. The reason that explains a higher IC_50_ for its counterpart **3f** (6.39 ± 0.46 μM) would be that the linker is connected to the 3-aminophenol O-atom and this moiety is stabilized by only one H-bond with Tyr354, while the 3-aminophenol fragment of compound **4f** is stabilized by two H-bonds, with Tyr354 and Glu434 provided by both hydrogens from –NH- and OH groups (Figure 3A,B). Nevertheless, antiproliferative effects must be explained taking into account not only the inhibition of the enzyme but also the clogP, a measure of lipophilicity of a compound. In accordance with these two parameters, the best antiproliferative activities are obtained with compounds endowed with high clog P (**3c**, **3d**, **4c**, **4d**), a consequence of a longer linker and two additional carbons in the pyrrolidine fragment. Interestingly, these compounds appear to have low toxicity as they have no significant effect on either quiescent or PHA-stimulated human lymphocytes. They also induce apoptosis in a dose dependent manner in Jurkat cells. Choline uptake assays also highlight the dual target of these compounds, where lipophilicity plays an essential role in the antitumor capacity of the compounds without ruling out that other mechanisms may contribute to antiproliferative SARs.

In summary, in this work, we present a series of monocationic compounds derived from the previously published **5** and **6** inhibitors, in which the adenine fragment has been replaced by aminophenol. The result has been an outstanding increase in antiproliferative potency up to nanomolar ranges. In addition, the compounds are also more active than the previously developed biscationic compounds. Moreover, the fact that they have only a positive charge makes them more lipophilic than the bis-cationic ones. That could facilitate reaching the enzyme, making them suitable candidates for hypothetical nanotransport.

## Figures and Tables

**Figure 1 pharmaceutics-13-01360-f001:**
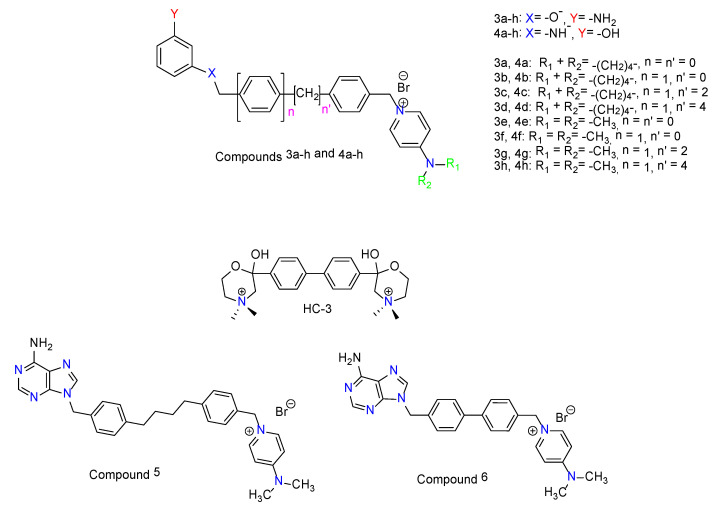
Structures of hemicolinium-3 (**HC-3**), the monocationic compounds **5** and **6**, and the previously published monocationic **3a**–**h** and **4a**–**h** structures. [19].

**Figure 2 pharmaceutics-13-01360-f002:**
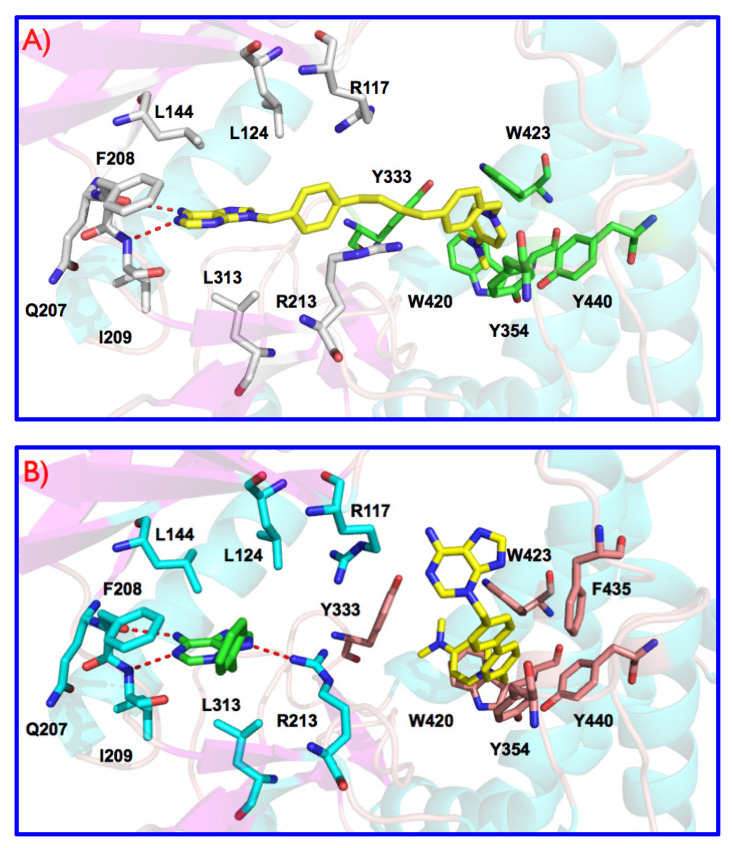
(**A**) Crystal structure of ChoK-α1/**5** complex (PDB ID: 3ZM9). Compound **5** (carbon atoms in yellow color) is inserted into ATP (carbon atoms in white color) and Cho (carbon atoms in green color) binding sites. (**B**) Crystal structure of ChoK-α1/**6** complex (PDB ID: 4BR3). One molecule of compound **6** (carbon atoms in green color) is inserted into the ATP binding site (carbon atoms in slate blue color) and another one (carbon atoms in yellow color) is inserted into the Cho binding site (carbon atoms in salmon color). Compound **5**: 1-[4-(4-{4-[(6-amino-9H-purin-9-yl)methyl]phenyl}butyl)benzyl]-4-(dimethylamino)pyridinium bromide. Compound **6**: 1-(4-{4-[(6-amino-3H-purin-3-yl)methyl]phenyl}benzyl)-4-(dimethylamino)pyridinium bromide.

**Figure 3 pharmaceutics-13-01360-f003:**
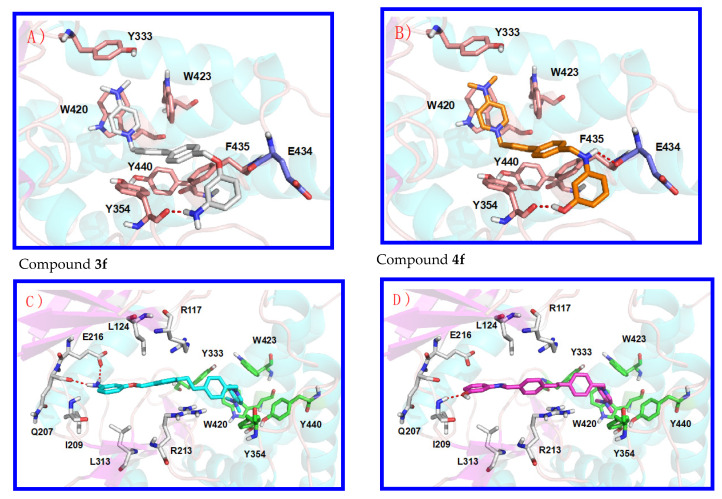
Obtained pose of compounds **3f** ((**A**), carbon atoms in white colors) and **4f** ((**B**), carbon atoms in orange colors) inserted into the de Cho binding site similarly to compound 6 (PDB ID: 4BR3), and compounds **3g** ((**C**), carbon atoms in cyan colors) and **4g** ((**D**), carbon atoms in magenta colors) inserted in both ATP and Cho binding sites similarly to compound **5** ((**D**), PDB ID: 3ZM9).

**Figure 4 pharmaceutics-13-01360-f004:**
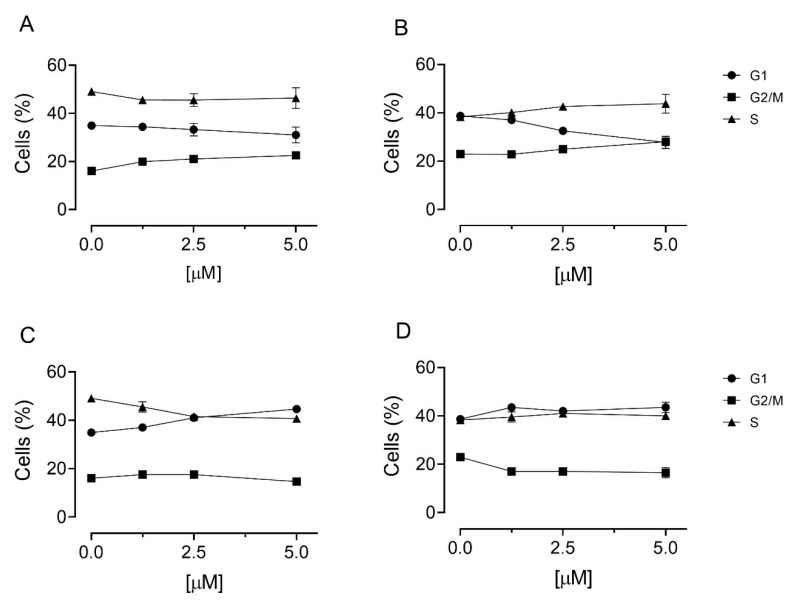
Effect of **4f** (**A**,**B**) and **3c** (**C**,**D**) on cell cycle in Jurkat cells. Cells were treated with the compounds for 24 h (**A**,**C**) and 48 h (**B**,**D**), at the concentrations of 1.25, 2.5, and 5.0 µM. Cells were then fixed and labeled with PI and analyzed by flow cytometry, as described in the experimental section. Data are represented as the mean of two separate experiments ± SEM.

**Figure 5 pharmaceutics-13-01360-f005:**
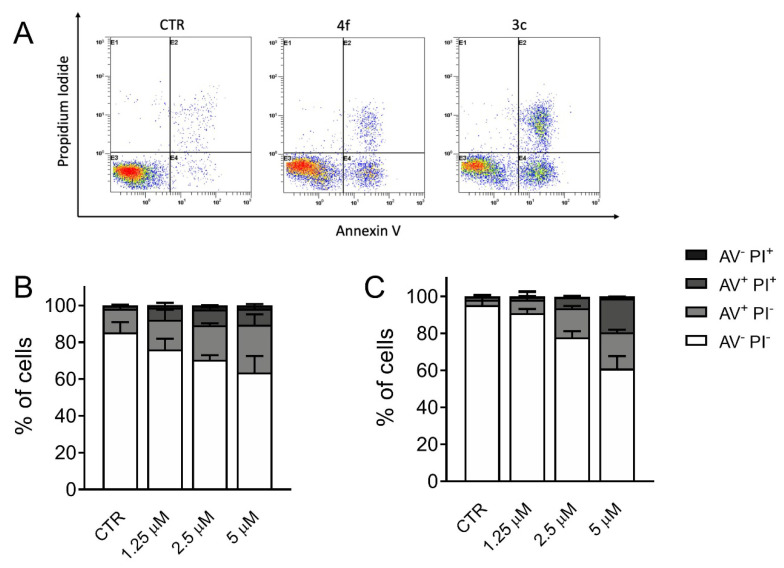
Panel A. Representative histograms of flow cytometric analysis of apoptotic cells after treatment of Jurkat cells with compounds **4f** and **3c** (**A**) of the indicated compounds at the concentration of 5 µM. Quantitative analysis of compounds **4f** (**B**) and **3c** (**C**) at the indicated concentrations after incubation for 48 h. The cells were then harvested and labeled with annexin-V-FITC and PI and analyzed by flow cytometry. Data are represented as mean ± SEM of three independent experiments.

**Table 1 pharmaceutics-13-01360-t001:** In vitro inhibitory effects of compounds **3a**–**h** and **4a**–**h**.

Comp.	clogP ^a^	IC_50_ ^b^ (μM) *h*ChoKα1	GI_50_ ^c^ (μM)					
			HT-29	HeLa	MCF-7	CCRF-CEM	HL-60	RS4;11
**3a**	0.68	>50	36.0 ± 7.4	4.8 ± 0.58	22.8 ± 4.3	3.4 ± 0.7	4.8 ± 0.4	16.8 ± 4.8
**3b**	2.57	17.55 ± 1.24	nd	nd	nd	nd	nd	nd
**3c**	3.13	8.56 ± 0.94	0.63 ± 0.14	0.068 ± 0.028	0.3 ± 0.1	0.03 ± 0.003	0.011 ± 0.005	0.15 ± 0.012
**3d**	4.18	6.79 ± 0.47	0.58 ± 0.17	0.16 ± 0.048	0.5 ± 0.2	0.015 ± 0.005	0.12 ± 0.091	0.29 ± 0.053
**3e**	0.57	6.33 ± 0.80	26.1 ± 6.7	8.2 ± 2.4	23.6 ± 5.1	1.2 ± 0.5	38.7 ± 6.2	53.6 ± 13.4
**3f**	2.45	6.39 ± 0.46	4.4 ± 0.7	4.2 ± 0.5	7.6 ± 1.6	0.54 ± 0.21	2.4 ± 0.7	2.0 ± 0.5
**3g**	3.01	4.29± 0.12	2.2 ± 0.9	1.2 ± 0.2	4.3 ± 0.9	0.40 ± 0.26	1.6 ± 0.9	2.6 ± 0.2
**3h**	4.07	7.01 ± 0.13	nd	nd	nd	nd	nd	nd
**4a**	0.33	3.34 ± 0.34	49.7 ± 11.4	12.8 ± 2.6	17.8 ± 5.3	7.3 ± 2.5	4.1 ± 1.2	24.5 ± 2.9
**4b**	2.22	3.85 ± 0.10	6.6 ± 2.3	2.1 ± 0.4	4.4 ± 0.5	0.5 ± 0.1	1.1 ± 0.3	2.8 ± 0.6
**4c**	2.78	3.83 ± 0.91	5.3 ± 2.1	0.86 ± 0.26	1.7 ± 0.4	0.14 ± 0.03	0.065 ± 0.023	0.46 ± 0.061
**4d**	3.84	3.36 ± 0.17	5.8 ± 1.9	0.39 ± 0.12	2.8 ± 0.5	0.086 ± 0.03	0.33 ± 0.074	0.71 ± 0.16
**4e**	0.22	2.82 ±0.16	23.8 ± 8.2	12.42 ± 1.8	6.7 ± 1.5	8.4 ± 1.6	4.0 ± 0.9	31.0 ± 2.6
**4f**	2.10	0,99 ± 0,17	13.2 ± 5.0	3.6 ± 0.4	5.0 ± 1.0	1.1 ± 0.4	1.3 ± 0.4	3.7 ± 0.4
**4g**	2.66	3.66 ± 0.11	5.2 ± 1.7	1.7 ± 0.2	2.4 ± 0.2	0.7 ± 0.1	0.25 ± 0.066	1.1 ± 0.3
**4h**	3.72	6.39 ± 0.46	3.6 ± 1.4	0.7 ± 0.1	2.2 ± 1.0	0.16 ± 0.034	0.37 ± 0.079	0.5 ± 0.08

^a^ The values of clogP were calculated with Chemdraw 15.0. ^b^ IC_50_: Compound concentration required to inhibit ChoKα1 enzyme by 50%; ^c^ GI_50_: Compound concentration required to inhibit tumor cell proliferation by 50%; nd: Not determined.

**Table 2 pharmaceutics-13-01360-t002:** Cytotoxicity of compounds **3c** and **4f**, for human peripheral blood lymphocytes (PBL).

	GI_50_ (µM) ^a^	
	3c	4f
PBL_resting_ ^b^	>10	7.6 ± 1.1
PBL_PHA_ ^c^	>10	3.6 ± 0.8

^a^ Compound concentration required to inhibit cell growth by 50%.^b^ PBL not stimulated with PHA. ^c^ PBL stimulated with PHA. Values are the mean ± SEM for three separate experiments.

**Table 3 pharmaceutics-13-01360-t003:** IC_50_ (μM) inhibition of [3H]choline uptake by compounds **3d** and **4f** in HepG2 cells.

Compounds	IC_50_ (Choline Uptake, μM)
3d	15.8 ± 0.24
4f	3.5 ± 0.07

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
