# Peer review of "Anticancer and Structure Activity Relationship of Non-Symmetrical Choline Kinase Inhibitors"

_pharmaceutics, 2021, doi:10.3390/pharmaceutics13091360_

Round 1
Reviewer 1 Report
Manuscript entitled “Anticancer Activity and Structure Activity Relationship of 2
Non-Symmetrical Choline Kinase Inhibitors” is a very interesting paper. However some suggestions need to follow before being considered for publication:
- The title of manuscript should be modified, remove one "activity" and adjust it.
- In general, English language and style must be improve in the whole manuscript,
- Please modify graphical abstract by giving some examples that these inhibitors are used for which kind of cancers
-
In the introduction, the background about the cancer is little, the authors should enrich this part by citing some recent literatures and emphasize the necessity of the cancer: https://pubs.rsc.org/en/content/articlelanding/2019/bm/c9bm00139e/unauth , https://pubs.rsc.org/en/content/articlelanding/2019/tb/c9tb01842e/unauth , https://onlinelibrary.wiley.com/doi/abs/10.1002/adhm.202002081
- References should be checked carefully
- 2.3 Please give detail methodology in this section about Docking study
- Why authors used two different kind of cancer cell lines i.e Hela and MCF-7?
- In materials section, authors need to clearly add on section about cell lines and protocols.
- What is effect of these inhibitors on normal cell lines? is it safe for normal cells
- 3.5. Measurement of apoptosis by flow cytometry: Please also put apoptosis flow cytometry results obtain from instrument.
- Discussion:This part requires a thorough development. The authors should clarify the signalized doubts. They should to demonstrate the advantages and disadvantages of the proposed inhibitors.At same time also compare results with other literatures.
Author Response
Non-Symmetrical Choline Kinase Inhibitors” is a very interesting paper. However some suggestions need to follow before being considered for publication:
The title of manuscript should be modified, remove one "activity" and adjust it.
Thank you for your comments; we have removed one "activity" and adjust it.
In general, English language and style must be improve in the whole manuscript,
Thank you for your comments. We have revised the whole version.
Please modify graphical abstract by giving some examples that these inhibitors are used for which kind of cancers
Thank you for your comments; we have changed the graphical abstract for a more clarifying one.
In the introduction, the background about the cancer is little, the authors should enrich this part by citing some recent literatures and emphasize the necessity of the cancer: https://pubs.rsc.org/en/content/articlelanding/2019/bm/c9bm00139e/unauth, https://pubs.rsc.org/en/content/articlelanding/2019/tb/c9tb01842e/unauth, https://onlinelibrary.wiley.com/doi/abs/10.1002/adhm.202002081, https://pubs.rsc.org/en/content/articlelanding/2021/bm/d1bm00721a/unauth, https://www.sciencedirect.com/science/article/abs/pii/S016773222101789X
References should be checked carefully
Thank you for your comments, we have included a detailed comment in the Introduction highlighting the importance of new therapeutic strategies and have referenced some of the very interesting articles recommended. We have also carefully reviewed the references.
2.3 Please give detail methodology in this section about Docking study
Thank you for your comments; we have include methodology in this section about docking study (paragraph 2.3).
Why authors used two different kind of cancer cell lines i.e Hela and MCF-7?
We actually used to evaluate the ability to induce apoptosis of derivatives 3c and 4f a T-leukemia cell line (Jurkat). In our previous paper we demonstrated (Ref 16 in this manuscript) as the T-lymphoblastic leukemia lines overexpressed Chokα and are particularly sensitive to Choline kinase inhibitors. An appropriate sentence to justify the choice of Jurkat cells has been inserted into the text
In materials section, authors need to clearly add on section about cell lines and protocols.
Accordingly we have inserted from paragraph 2.4 to 2.8 the detailed protocols of the methods used together with the description of the cell lines.
What is effect of these inhibitors on normal cell lines? is it safe for normal cells
Thank you for your comments, however we had already included in section 3.3. Effects of compound 3c and 4f in non-tumor cells, showing that these derivatives appear endowed with low toxicity in human peripheral blood lymphocytes (PBL).
3.5. Measurement of apoptosis by flow cytometry: Please also put apoptosis flow cytometry results obtain from instrument.
In accordance with the Reviewer's suggestions, we have inserted in Figure 4 as panel A the image of three representative histograms relating to the flow cytometric analysis performed for the two compounds at a concentration of 5 µM. The quantitative analysis deriving from the histograms is shown in the same figure as panels B and C for compounds 4f and 3c respectively.
Discussion:This part requires a thorough development. The authors should clarify the signalized doubts. They should to demonstrate the advantages and disadvantages of the proposed inhibitors. At same time also compare results with other literatures.
Thank you for your comments; we have introduced a detailed comment in the conclusions where we explain the advantages of these new compounds.

Reviewer 2 Report
I found the manuscript very confusing. If my understanding is correct, the authors’ main aim was to re-evaluate the activity of a series of choline kinase inhibitors that they previously synthesized and characterized (mostly in reference [15]), using a pure recombinant enzyme that was not available to them previously. Unexpectedly, one of the compounds (named 4f) that did not show an outstanding activity in [15], was highly active in this study. In addition, the authors have tested the cytotoxicity of their previously reported compounds in a panel of cancer cell lines and found that two of the compounds (3c and 3d) were highly cytotoxic with IC50 values in the sub-micromolar range. To explain these results, the authors performed molecular docking studies with the recently published crystal structures of human choline kinases (PDB ID 4BR3 and 3ZM9). These results are interesting enough to warrant a publication, but the aims, results and conclusions should be presented in a more straightforward manner. For instance, there is no need for a long introduction that explains the importance of choline phosphorylation and includes a scheme (Scheme 1), since there is a plenty of published reviews on the topic. The numbering of the compounds 3a-h and 4a-h in Figure 1 was apparently taken from [15], but in the context of the current work, it is unclear why compounds 5 and 6 appear in Figure 1 before 3 and 4, and no compounds 1 and 2 are mentioned. It would also be useful to show the structures of the most active compounds found in this study (4f and 3c,d) separately, to avoid misinterpretation. In addition, the resolution of Figure 1 should be improved. In Table 1, it is unclear what the compounds MN-48b and RSM-932A refer to (their structures should be shown in Figure 1). It is also unclear why the activities of the studied compounds were not compared with that of the reference compound, HC-3. As a minor point, it is unclear why some of the reference numbers are highlighted in red, while the others are not. The English of the manuscript will require some editing, particularly with regards to punctuation.
Author Response
Reviewer 2
I found the manuscript very confusing. If my understanding is correct, the authors’ main aim was to re-evaluate the activity of a series of choline kinase inhibitors that they previously synthesized and characterized (mostly in reference [15]), using a pure recombinant enzyme that was not available to them previously. Unexpectedly, one of the compounds (named 4f) that did not show an outstanding activity in [15], was highly active in this study. In addition, the authors have tested the cytotoxicity of their previously reported compounds in a panel of cancer cell lines and found that two of the compounds (3c and 3d) were highly cytotoxic with IC50 values in the sub-micromolar range. To explain these results, the authors performed molecular docking studies with the recently published crystal structures of human choline kinases (PDB ID 4BR3 and 3ZM9). These results are interesting enough to warrant a publication, but the aims, results and conclusions should be presented in a more straightforward manner. For instance, there is no need for a long introduction that explains the importance of choline phosphorylation and includes a scheme (Scheme 1), since there is a plenty of published reviews on the topic. The numbering of the compounds 3a-h and 4a-h in Figure 1 was apparently taken from [15], but in the context of the current work, it is unclear why compounds 5 and 6 appear in Figure 1 before 3 and 4, and no compounds 1 and 2 are mentioned. It would also be useful to show the structures of the most active compounds found in this study (4f and 3c,d) separately, to avoid misinterpretation. In addition, the resolution of Figure 1 should be improved. In Table 1, it is unclear what the compounds MN-48b and RSM-932A refer to (their structures should be shown in Figure 1). It is also unclear why the activities of the studied compounds were not compared with that of the reference compound, HC-3. As a minor point, it is unclear why some of the reference numbers are highlighted in red, while the others are not. The English of the manuscript will require some editing, particularly with regards to punctuation.
Thank you for your constructive comments:
These results are interesting enough to warrant a publication, but the aims, results and conclusions should be presented in a more straightforward manner. For instance, there is no need for a long introduction that explains the importance of choline phosphorylation and includes a scheme (Scheme 1), since there is a plenty of published reviews on the topic.
Thank you for your comments. Since this article belongs to a special chapter on choline kinase, we consider it necessary to elaborate on the introduction about the importance of choline phosphorylation. However, we have removed scheme 1. We have introduced detailed comments in introductions and conclusions sections where we explain the advantages of these new compounds.
The numbering of the compounds 3a-h and 4a-h in Figure 1 was apparently taken from [15], but in the context of the current work, it is unclear why compounds 5 and 6 appear in Figure 1 before 3 and 4, and no compounds 1 and 2 are mentioned.
We name the compounds as 5 and 6 as the numeration following that of the final compounds, instead of 1 and 2 as previously presented. However, for the better understanding of the reader we have moved the position of these compounds in the Figure 1.
It would also be useful to show the structures of the most active compounds found in this study (4f and 3c,d) separately, to avoid misinterpretation.
Thank you for your comments. We have included in the new graphical abstract the structure of the most interesting compounds 3c,d and 4f.
In Table 1, it is unclear what the compounds MN-48b and RSM-932A refer to (their structures should be shown in Figure 1). It is also unclear why the activities of the studied compounds were not compared with that of the reference compound, HC-3.
Thank you for your comments. Compounds HC-3, 5 and 6 are not included in Table 1 because they were not tested in the same cell lines, but they were only used for design and docking studies. On the other hand, to avoid misinterpretations we have removed compounds MN48b and RSM932A from the table
As a minor point, it is unclear why some of the reference numbers are highlighted in red, while the others are not.
Thank you for your comments. We have highlighted in red (now in new version in blue) all the references just for better understanding of the reviewers, the final version would be in black.
The English of the manuscript will require some editing, particularly with regards to punctuation.
Thank you for your comments. We have revised the whole version.

Round 2
Reviewer 2 Report
The manuscript has improved significantly after the revision. My main suggestion is that the Abstract, Introduction and Conclusion should state clearly that this work presents re-evaluation of known compounds, rather than synthesis and characterization of new compounds. For instance, the mention of ‘additional library of compounds’ (last sentence before Figure 1) is misleading. In my copy of the manuscript, Figure 3 shows empty rectangles instead of structures. This needs to be checked.
Author Response
Reviewer 2
The manuscript has improved significantly after the revision. My main suggestion is that the Abstract, Introduction and Conclusion should state clearly that this work presents re-evaluation of known compounds, rather than synthesis and characterization of new compounds. For instance, the mention of ‘additional library of compounds’ (last sentence before Figure 1) is misleading. In my copy of the manuscript, Figure 3 shows empty rectangles instead of structures. This needs to be checked.
Thank you for your constructive comments.
We have introduced in the summary, introduction and conclusions some sentences that clarify that this work refers to more in-depth studies of previously published compounds. We have also introduced the complete figure 3.
